

# Durability and physical characterization of anti-fogging solution for 3D-printed clear masks and face shields

Succhay Gadhar*, Shaina Chechang*, Philip Sales and Praveen Arany

Oral Biology, Biomedical Engineering, and Surgery, University at Buffalo, Buffalo, NY, USA
* These authors contributed equally to this work.

## ABSTRACT

**Background:** The COVID-19 pandemic brought forth the crucial roles of personal protective equipment (PPE) such as face masks and shields. Additive manufacturing with 3D printing enabled customization and generation of transparent PPEs. However, these devices were prone to condensation from normal breathing. This study was motivated to seek a safe, non-toxic, and durable anti-fogging solution.
**Methods:** We used additive 3D printing to generate the testing apparatus for contact angle, sliding angle, and surface contact testing. We examined several formulations of carnauba wax to beeswax in different solvents and spray-coated them on PETG transparent sheets to test contact and sliding angle, and transmittance. Further, the integrity of this surface following several disinfection methods such as detergent, isopropyl alcohol, or water alone with gauze, paper towels, and microfiber, along with disinfectant wipes, was assessed.
**Results:** The results indicate a 1:2 ratio of carnauba to beeswax in Acetone optimally generated a highly hydrophobic surface (contact angle $150.3 \pm 2.1°$ and sliding angle $13.7 \pm 2.1°$) with maximal transmittance. The use of detergent for disinfection resulted in the complete removal of the anti-fogging coating, while isopropyl alcohol and gauze optimally maintained the integrity of the coated surface. Finally, the contact surface testing apparatus generated a light touch ($5,000 \ N/m^2$) that demonstrated good integrity of the antifogging surface.
**Conclusions:** This study demonstrates that a simple natural wax hydrophobic formulation can serve as a safe, non-toxic, and sustainable anti-fogging coating for clear PPEs compared to several commercial solutions.

## INTRODUCTION

The COVID-19 pandemic has had a significant toll on the public health, biomedical, and financial aspects that have significantly changed society (*Pak et al., 2020*; *Pirasteh-Anosheh et al., 2021*). Among various healthcare services, the presence of SARS-CoV in the nasal and upper respiratory tract presented significant challenges for dentists and anesthetists. Clinical dentistry presented a significant challenge due to the proximity and inherent aerosol-generating procedures (*Goriuc et al., 2022*; *Kurian, Gandhi & Thomas, 2021*). The use of personal protection equipment (PPE) such as face masks and face shields was

Corresponding author
Praveen Arany,
praveenarany@gmail.com

integral in reducing the spread of infections among the general population (*Artenstein, 2020*). While dentistry routinely utilizes face masks, double masking, and the highly protective N95, the shielding nature and the lack of face-to face exposure further compounded the pandemic fear and anxiety in patients naturally apprehensive about their dental visits (*Marcus et al., 2022*; *Muneer et al., 2022*; *Nikolic et al., 2022*). Another major limitation of these opaque masks is that they hinder effective communication due to the lack of non-verbal cues such as facial expressions and lips movements (*Blais et al., 2017*; *Chodosh, Weinstein & Blustein, 2020*; *Chu et al., 2021*). This is particularly challenging in patients with differently-abled hearing due to increased signal-to-noise from the ambient noise in dental operatories (*Fleming, Maddox & Shinn-Cunningham, 2021*; *MacLeod & Summerfield, 1987*; *Schwartz, Berthommier & Savariaux, 2004*; *Sonnichsen et al., 2022*). The use of transparent masks and face shields has been demonstrated to improve these limitations (*Atcherson et al., 2017*).

Another major challenge during the early phase of the pandemic was the significant disruption of the PPE supply chain requiring local innovations where additive manufacturing using 3D printing came to the forefront (*Belhouideg, 2020*; *O'Connor et al., 2022*; *Rupesh Kumar et al., 2022*; *Vallatos et al., 2021*). The ability to generate required quantities based on need and additional customization for individual applications were additional significant advantages of this approach (*Kalkal et al., 2022*). Building on this, in collaboration with e-NABLE, a 3D printing community, students at the University at Buffalo School of Dental Medicine generated transparent PPEs using 3D printing and vacuum casting. Among them, several custom designs were specifically generated to accommodate commercially available dental loupes that are made available freely online (https://www.buffalo3dppe.com/). These clear masks and face shields were well accepted and generated much enthusiasm among clinical users. Besides dentistry, another group of clinicians that notably adopted these clear PPEs were the local speech and hearing-impaired clinics. However, an immediate limitation of this approach reported by users was the fogging due to the moisture from regular breathing. This essentially negated the advantage of the transparent nature of these masks and face shields and presented additional hindrance of needing repetitive wipe-downs.

The commercially available anti-fogging solutions used for eyeglasses and car exteriors contain polydimethylsiloxane and silica nanoparticles (*Wahab et al., 2023*). The long-term use of these solutions was considered unsuitable due to their potential for skin irritation, and potential inhalation or ingestion. Hence, we sought to identify a non-toxic, antifogging coating for the transparent masks and face shields for long-term use. To design such a solution, specific design criteria needed to be met. The surface of a water droplet is attracted to the bulk of the droplet due to cohesion which results in their beading. The shape that a drop takes on a given surface depends on the surface tension of the fluid and the physical characteristics of the surface. A surface-fluid interface is the contact angle that measures the wettability of a surface that can vary from hydrophilic (less than or equal to 30°) to hydrophobic (greater than or equal to 120°), where the higher the contact angle, the lower its wettability. Another measure of the extent of moisture retention on a surface is the sliding angle which is measured by tilting the surface with the solution bead to
determine the angle at which it rolls off. The more hydrophobic the surface, the higher the angle of loss of fluid droplets (*Lv et al., 2010*). Another key design aspect of the anti-fogging solution would be its resistance to wear due to repeated skin surface contact during mask use. The ideal solution would offer a thin, durable coating for daily use that would not need repetitive applications while maintaining maximal transparency.

Natural products offer attractive non-toxic coatings. Prior studies have examined the combination of cinnamon and nutmeg that produces high hydrophobicity (*Razavi et al., 2017*). Although both are natural products, this article utilized a organosilane-based alkyl and perfluorinated synthetic chemical coatings that raised some toxicity concerns due to high cuprous oxide content. We drew inspiration from the naturally occurring, extremely hydrophobic lotus leaves. These leaves expel water easily due to their micro-structural features, termed papillae, that exude epicuticular waxes, cutin (*Bormashenko, 2013*; *Li, 2020*). Alternatively, Carnauba and beeswax have also high hydrophobicity that are routinely used in the food service industry (*Wang & Shen, 2018*). With their high hydrophobicity, these solutions on food containers prevent waste by ensuring minimal food retention. However, there was a lack of clarity in the concentrations used in these prior formulations and the resulting hydrophobicity. The present study examined the different formulations of these two waxes together and evaluated the contact angle, sliding angle, and transmittance. The optimized formulation was subjected to durability testing with manual disinfectant procedures and long-term contact-wear testing.

## MATERIALS AND METHODS

### Formulation of anti-fogging solutions

Hydrophobic wax coatings were generated in acetone or methanol as solvents using ultrasonication. Different ratios of carnauba wax to beeswax (both Sigma Aldrich, St. Louis, MO, USA) were employed namely 0.4375 g:0.4375 g (1:1), 0.35 g:0.525 g (1:1.5), 0.33 g:0.66 g (1:2), 0.417 g:0.834 g (1:2 HC) were melted in a 50 ml plastic tube (Corning, Thermofisher, Waltham, MA, USA) in a water bath. After the waxes had melted, 25 milliliters of either acetone or methanol (both Sigma Aldrich, St. Louis, MO, USA) were added, and the solution was emulsified immediately using a probe ultrasonication (Q2000; QSonica, Newtown CT, USA) at 90% amplitude for 3 min and transferred into a 50 ml glass spray bottle.

### Contact and sliding angle testing

Solutions were sprayed onto a 2 × 2-inch sheet of PETG at roughly a distance of 5 inches, spraying 10 times. After drying, the sheets of PETG were tested for contact angle, sliding angle, and transmission. A custom contact and sliding angle device were generated based on commercially available models using online CAD software (Onshape; PTC, Rockwell Automation, Boston, MA, USA). The devices were printed using Polylactic acid (PLA) filament (Overture; Overture 3D Technologies, Missouri City, TX, USA) on a i3 Prusa 3D printer (Prusa Research, Prague, Czech Republic) (Figs. 1A–1C). The device included a slot for to hold a mobile phone to take digital pictures of the droplet. After fixing the plastic surface to the platform, droplets of water were generated with a 3 ml syringe and 14-gauge

needle (to mimic humidity post-condensation on clear PPEs) and digital image were captured for analysis (Fig. 1D). The digital images were analyzed using the NIH ImageJ (ver. 1.53n) software. The sliding angle set up included a protractor to assess the angle at which the droplet slides off. The platform was tilted slowly until the water droplet slid off, and the angle was documented (Fig. 1E). All studies were performed in performed in triplicate and repeated atleast twice.

## Transmittance analysis

The laser apparatus consisted of a 650 nm diode laser (Weber Medical, Beverungen, Germany) at 10 mW/cm$^2$, and a sensor with power meter (both Thor Labs, Newton, NJ, USA) (Fig. 1F). The sheets of PETG were placed on top of the power sensor, and transmission was assessed.

## Optical clarity analysis

A sticker was placed on a bench and the sheets of PETG were placed over it. Digital pictures were taken with a mobile phone camera. The coated sheets of PETG were compared with the uncoated control and the visual clarity of the coatings was assessed.

## Routine manual disinfection testing

The PETG coated surfaces were subjected to different solutions such as soap solution (10% v/v, Dawn; Procter & Gamble, Cincinnati, OH, USA), isopropyl alcohol (70%, Sigma Aldrich, St. Louis, MO, USA), and water alone using paper towels (Uline, Milton, ON, USA), gauge (Henry Schein, New York, NY, USA) or microfibers (Magic Fiber, Miami, FL, USA). These disinfectants were applied ten times in a uniform, lateral motion with manual pressure by a single operator. Disinfection wipes (Metrex, Orange, CA, USA) were used in another group.

## Coating durability following contact wear testing

To simulate accelerated contact wear testing, a custom apparatus was designed to mimic skin contact. Using the CAD software (Onshape; PTC, Rockwell Automation, Boston, MA, USA), a rotating platform with spring-loaded bases was designed to apply suitable skin contact forces (Brill et al., 2018; Dellweg et al., 2010; Schettino et al., 2001). The device was 3D printed with PLA filament (Overture; Overture 3D Technologies, Missouri City, TX, USA) on a i3 Prusa 3D printer (Prusa Research, Prague, Czech Republic). The wheel was connected to a DC Motor (Greartisan; Shenzen Hotec, Shenzhen, China) via a speed controller and a 9 V battery which allowed the motor to function up to 1,000 RPMs. After the coated samples were mounted on the platforms, the apparatus was positioned to allow the platforms to extend and retract using centrifugal force while spinning, hitting a skin-equivalent surface (rough side of oil-treated leather) to simulate contact. To simulate multiple skin contacts, the device was operated for 5 min, allowing each platform to contact the wall about 5,000 times for each sample to simulate 4 weeks of routine wear. The contact force applied by the rotational, spring-loaded platform was measured with a digital force probe (Baoshishan, China). Following the contact wear testing, samples were

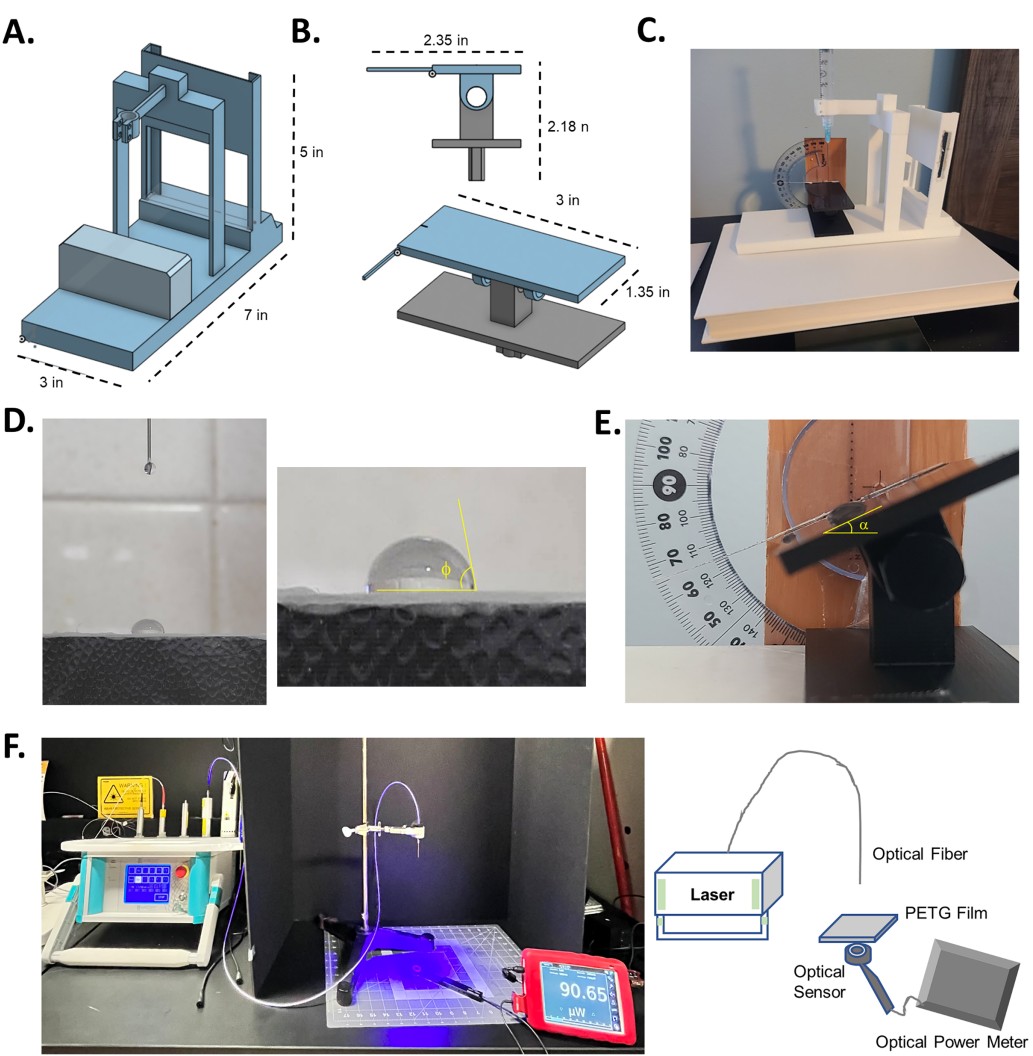

**Figure 1 Physical characterization of anti-fogging coating.** (A) CAD of the contact angle apparatus for 3D printing. (B) CAD of the sliding angle module for 3D printing. (C) 3D printed models of contact and sliding angle testing apparatus. (D) Image of the contact angle assessment showing droplet generation and analysis. Inset shows the digital analysis of contact angle φ using NIH ImageJ. (E) Digital image analysis showing the sliding angle measurement using NIH ImageJ. (F) Transmittance set-up using a diode laser and optical power meter with the sensor.

assessed for changes in contact angle, sliding angle, transmittance and examined with topological analysis.

## Topological analysis

Scanning electron microscopy (SEM) analysis was performed to determine the topology and the depth of the coating on the plastic sheet samples prior to and following wear testing. Samples were sputter-coated (Cressington 108; Ted Pella Inc, Redding, CA, USA) with carbon for 120 s and examined using SEM (S-4700; Hitachi, Tokyo, Japan) with a voltage of 2.0 kV at about 14.5 ± 0.4 mm using secondary electron mode.

## Statistical analysis

Data was organized in Excel (Microsoft, Seattle, WA, USA) and presented as Means with standard deviations. Data was subjected to a Student's T test or two-way analysis of variance (ANOVA) among different treatments with Bonferroni's correction for multiple comparisons where appropriate. A $p < 0.05$ was considered statistically significant.

# RESULTS

## Formulation and characterization of anti-fogging solutions

We first examined different ratios of carnauba and beeswax in two solvents, namely acetone and ethanol, to generate a homogenous solution (Fig. 2A). The contact angles of the acetone-based solutions were higher than their methanol counterparts (Figs. 2B and 2C). The coating with the highest contact angle was the 1:2 ratio in acetone solution with a mean contact angle of 150.3 ± 2.1°. This categorizes this surface coating as highly hydrophobic compared to the other coatings and was significantly ($p < 0.05$) more hydrophobic than the uncoated plastic surface. We also examined the higher concentrations (HC) at 1:2 ratio to inquire if the density and hence, eventual coating concentration would affect these properties, but did not observe any further significantly improved characteristics. As expected the sliding angle for this coating was the lowest at 13.7 ± 2.1° which was significantly ($p < 0.05$) lower than the uncoated plastic surface (Figs. 2D and 2E). The transmittance of these formulation appears to vary with the concentration of the beeswax component as lower amounts resulted in higher transmittance (Figs. 2F and 2G). The optical clarity was also assessed through the observation of the sticker through the coated slides (Fig. 2H). The optical clarity decreased as the concentrations of the solutes increased as was expected. The methanol-based formulations had a comparable degree of transmittance (0.41 to 0.33) than the acetone (0.3 to 0.26) counterparts compared to control (uncoated 0.43 ± 0.02). Based on these results, we decided to pursue the 1:2 carnauba to bee wax in acetone formulation for subsequent studies.

## Effect of routine disinfection procedures on durability of antifogging solution

The use of disinfectants is a necessary part of routine plastic mask and face shield use due to the prominent aerosol generating dental procedures (*Epstein, Chow & Mathias, 2021*; *Mick & Murphy, 2020*; *Wilson et al., 2020*). Next, we examined routine disinfectant procedures employed in the dental office namely, soap, water, and 70% isopropyl alcohol with paper towels, gauze, or microfiber, and disinfectant wipes. We noted a significant ($p < 0.05$) reduction in the contact angle after all disinfection methods (Fig. 3A). Among all these methods, isopropyl alcohol with gauze reduced the contact angle the least (144.3 ± 1.6°) while soap made the surface most hydrophilic (wettable), removing the coating most effectively. Concurrently, the sliding angle analysis showed a similar trend, increasing overall with all, but the soap group, disinfection methods (Fig. 3B). The isopropyl alcohol group showed the least change and no statistically significant difference was noted when a paper towel was used. Interestingly, the transmittance appeared to vary after individual disinfectant methods with some showing increases while other showing significant
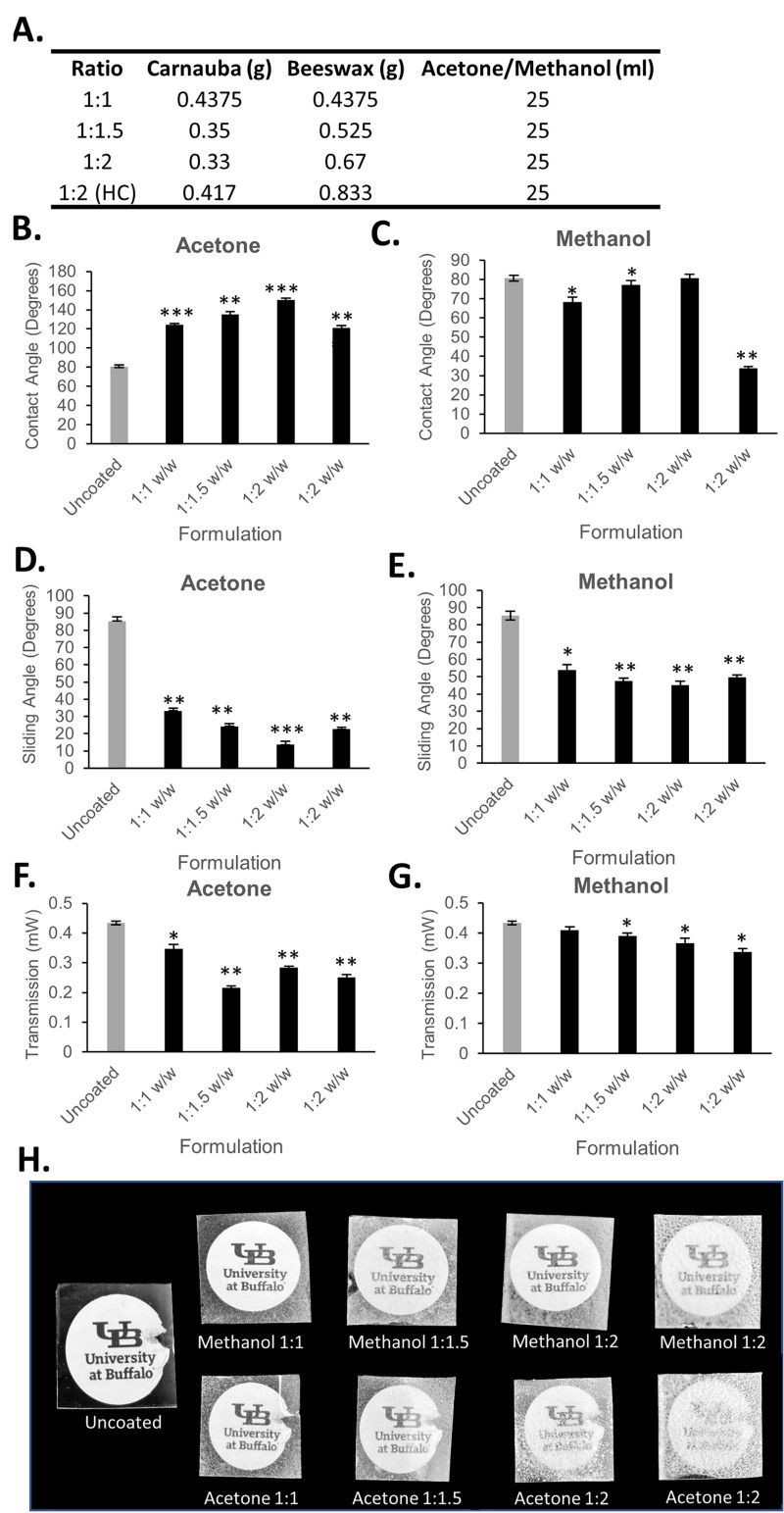

**Figure 2 Anti-fogging solution formulation and characterization.** (A) Tabular outline of the formulation depicting ratios of carnauba and beeswax in acetone and methanol used to coat PETG plastic sheets. (B and C) Contact angle assessment of Acetone and Methanol formulations at various concentrations. (D and E) Sliding angle analysis of Acetone and Methanol formulations at various concentrations. (F and G) Transmission of diode laser for transmittance assessment of Acetone and Methanol formulations at various concentrations. (H) Digital image of an uncoated,

**Figure 2** (continued)
Acetone and Methanol formulations applied to the PETG sheet. Data are shown as Mean and SD and representative of at least two independent studies. Statistical significance was determined using two-way analysis of variance (ANOVA) among different treatments using Bonferroni's multiple comparison test ($n$ = 3). Statistical significance is denoted as $^*p < 0.05$, $^{**}p < 0.005$, and $^{***}p < 0.0005$.               

decreases (Fig. 3C). The method of disinfection that showed most increase in light transmission were the sterilizing wipes by themselves. The lowest transmission was observed with isopropyl alcohol and soap with paper towels. Together, these results suggest that using isopropyl alcohol with gauze is optimal for disinfection as it minimally affects antifogging coating properties such as contact and sliding angle and transmission after repeated use.

### Antifogging coating stability after contact wear testing

Finally, we investigated the durability of the anti-fogging solutions as they would be subjected to rigorous daily wear and tear during PPE use. We first created a simple rig to simulate skin contact based on a 3D printed apparatus (Figs. 4A and 4B). The rig was designed as a rotating platform that would result in a repetitive contact of the coated device with a stationary leather surface. The pressure exerted by facemask on the nose bridge and chin contact has been assessed to be 45 to 91 mm of Hg (6,000–12,000 $N/m^2$) (Fig. 4C) (*Brill et al., 2018*; *Dellweg et al., 2010*; *Schettino et al., 2001*). The constant skin contact force in our device was assessed to be 5,000 $N/m^2$ that could be approximated to a repeated light skin contact.

Following the simulated wear, the contact and sliding angle as well as transmittance was assessed. We observed a significant reduction in contact angle (133.3 ± 1.5°) and concomitant increase in the sliding angle (23.7 ± 1.5°) compared to the uncoated control surfaces. While transmittance was not significantly different before and after wear simulation, there was a trend towards reduced transmission. Topological analysis noted the wear simulation resulted in significant accretions and scratches that likely contribute to the reduced light transmission observed. These results indicate that due to reduction in coating characteristics by about 50% over the simulated 4-week wear period, the coating will likely need to be replaced as often as every other week to maintain optimal functions.

## DISCUSSION

The COVID pandemic presented new challenges to healthcare with an urgent need for innovations in disinfection approaches. The use of PPE was central in the attempts to mitigate the spread of infection and a major part of the solution to mitigate the health crisis. The use of these barriers continues to have an impact on both the psychological, and medical well-being of both the patients and the healthcare professionals themselves (*Kisielinski et al., 2021*). The initial response to the pandemic brought manufacturing and supply chain disruptions resulting in additive 3D printing enthusiasts to offer custom, local solutions. Our team at the University at Buffalo focused on the transparent PPE designs

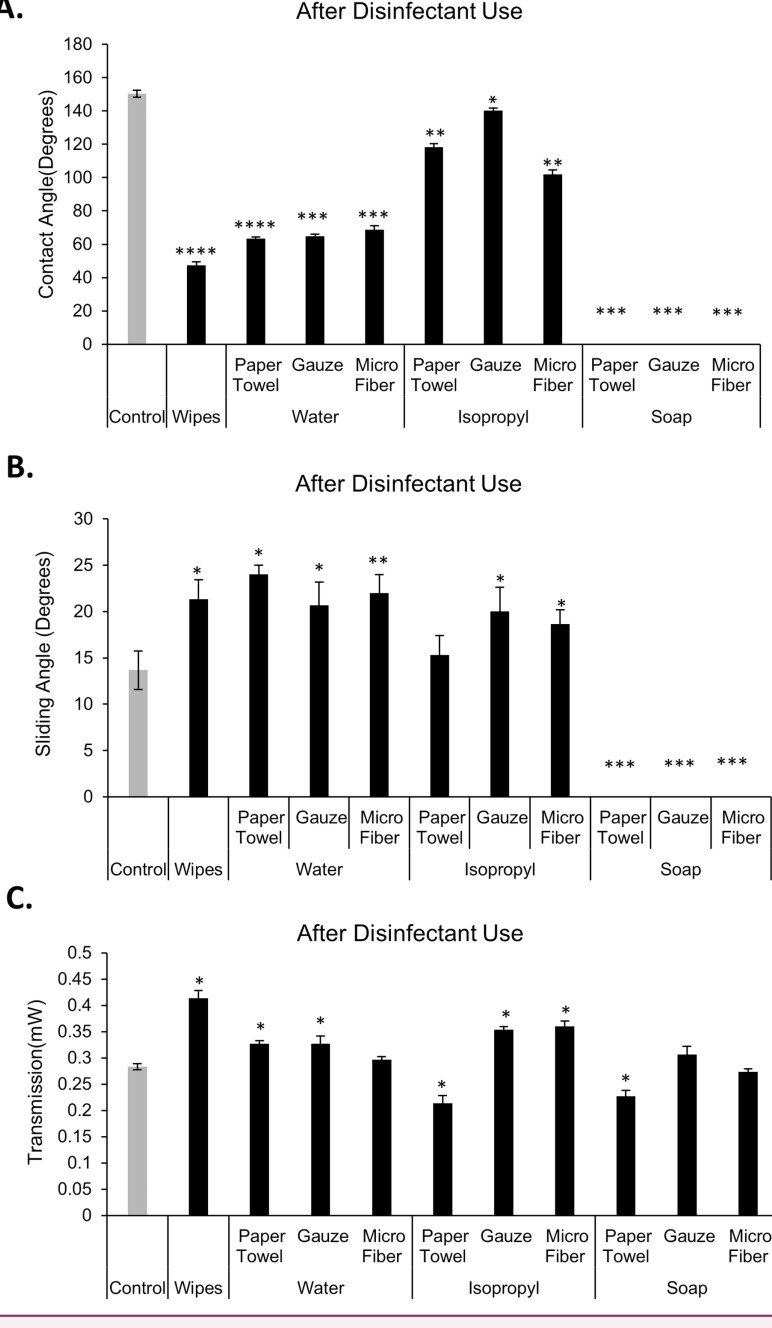

**Figure 3 Effects of disinfection on anti-fogging coating integrity.** (A) Contact angle assessment after various disinfection procedures. (B) Sliding angle analysis after various disinfection procedures. (C) Transmittance after various disinfection procedures. Data are shown as mean and SD and representative of at least two independent studies. Statistical significance was determined using two-way analysis of variance (ANOVA) among different treatments using Bonferroni's multiple comparison test ($n = 3$). Statistical significance is denoted as $^*p < 0.05$, $^{**}p < 0.005$, $^{***}p < 0.0005$, and $^{****}p < 0.00005$.

and dental loop attachments (Fig. 5). The condensation from breathing obscured the functions of the clear PPEs that presented a major impasse to their use. This work was specifically motivated to address this deterrent to routine clear PPE use.

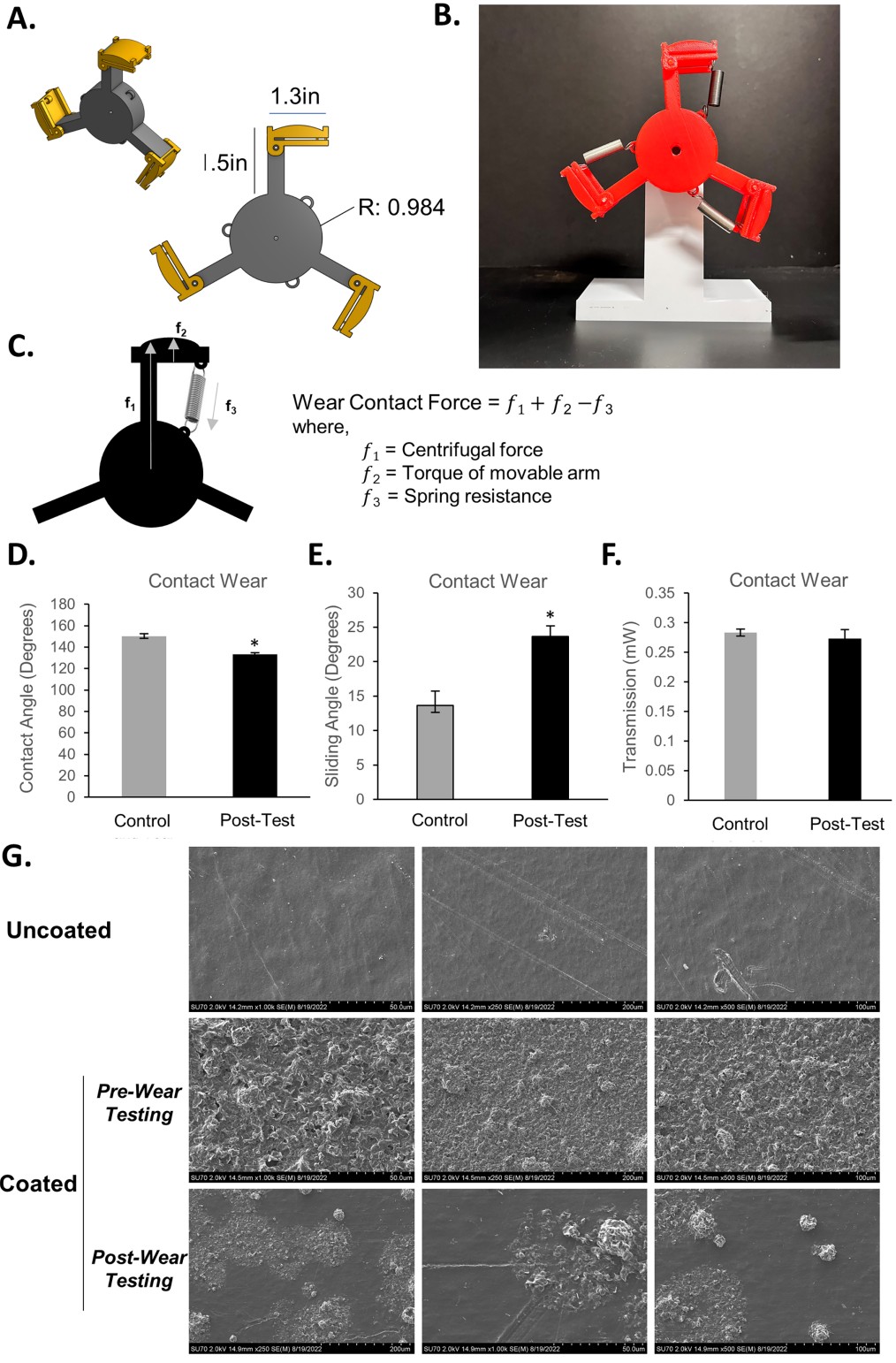

**Figure 4 Durability testing of anti-fogging coating following contact testing.** (A) CAD of contact testing apparatus used for wear testing analysis. (B) 3D printed model of the assembled contact testing apparatus. (C) Outline of the eminent forces determining the contact force on PETG surfaces. The contact angle (D), sliding angle (E), and transmittance (F) analysis before and after contact testing. (G) Scanning electron microscopy images of the uncoated and coated surfaces before and after contact

**Figure 4** (continued)
testing. Data are shown as Mean and SD and representative of at least two independent studies. Statistical significance was determined using the Students' T-test ($n = 3$). Statistical significance is denoted as *$p < 0.05$.

**A.**

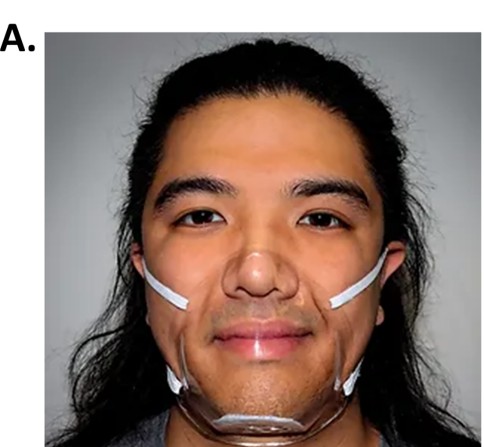

**B.**

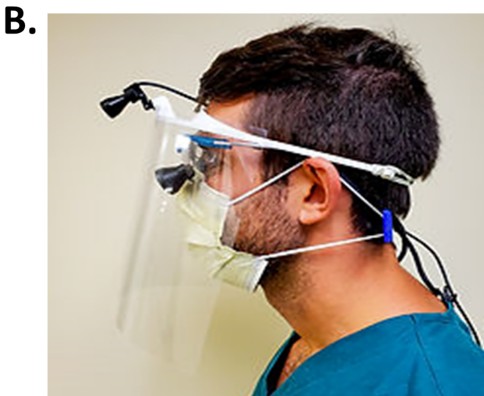

**C.**

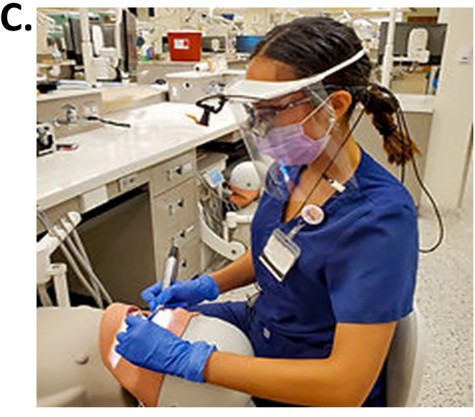

**Figure 5 Clear face shields and masks.** A clear mask design with filter unit in medium (A) and large (B) sizes. The 3D printed clear face shields are shown to accommodate different dental operating loupes such as eclipse (C), vision, surgitel and orascoptic. Images are provided with consent from lab volunteers from the website www.buffalo3dppe.com.

We tested polyethylene terephthalate glycol (PETG), a commonly-used thermoplastic with impact resistance, durability, ductility, chemical resistance properties well-suited to this application. To generate an anti-fogging solution for our clear face shields and masks, a major design challenge was the close proximity to nose and mouth (*Rothe et al., 2011*; *Thá et al., 2021*). We chose to pursue natural waxes as the main ingredient as they provided a natural, non-toxic solution that has a well-established track record of biological safety (*ACT, 1984*). The other major component constituent was a polar solvent to enhance miscibility and dispersion of these waxes. We chose to examine Acetone and Ethanol in our formulation. The acetone formulation was superior in generating the most hydrophobic product after spray coating. Our initial efforts at dissolving the waxes in Acetone (boiling point of 55.5 °C) by heating alone resulted in evaporation and non-uniform dissolution. Hence, we chose to disperse the waxes *via* ultrasonication. Among the various formulations, the most hydrophobic surface was generated by the 1:2 ratio (contact angle 150.3 ± 2.1°) though the other formulations were also strongly hydrophobic (contact angles 121° to 135°). We also examined Methanol (boiling point of 148.5 °F) in the same volume that, in contrast, generated hydrophilic surface coatings with all formulations (contact angles 34° to 82°), but most prominently with the 1:2 at the higher concentration (contact angle 80.7 ± 2.1°). Increasing the concentration in the Methanol formulation (1:2 HC) appeared to impact the physical characteristics more than the Acetone (1:2 HC) formulation that could be attributed to differential dissolution of the waxes and subsequent homogeneity of the coating film. Among the two solvents examined, Methanol did not enable a suitable anti-fogging formulation and hence, the 1:2 formulation in acetone was chosen as an optimal anti-fogging solution. This may have been due to the fact that methanol has a higher polarity than acetone that facilitates improved dispersibility of the non-polar wax compounds.

The use of surface disinfectants is an integral part of maintaining and reusing PPEs. This is a sustainable and cost-effective practice that is routinely employed. A key design requirement of the anti-fogging solution is durability with these disinfectant approaches. We observed that the soap solution group consistently demonstrated complete removal of the anti-fogging solution. While this desirable as a disinfectant approach to remove all potentially hazardous aerosols on these PPEs, its amphiphilic nature allowing it to form micelles with the wax components of the anti-fogging solution would deem it unsuitable. The removal of the anti-fogging solution with these approaches could be attributed to the surfactant nature that interferes with the surface tension of the water droplets. Another possibility is that the lipid-to-lipid interaction of the soap and the waxes may inactivate the hydrophobic nature of the coating. The superior solubilizing ability of isopropyl alcohol that may further smoothen the coating, reduce surface tension further or alter drying times that are areas of future investigations.

To investigate the durability of the anti-fogging coating, we generated a custom rig for accelerated surface contact testing. The major design principle for this device was to generate a PPE skin contact force. The force generated by our rig was 5,000 N/m$^2$ and was intended to approximate light skin contact force in non-fastened areas. This reflects routine surface contact during routine use of the PPEs. The hydrophobicity of the

anti-fogging surfaces was relatively well maintained (contact angle 133.3 ± 1.5° and sliding angle 23.7 ± 1.5°). However, the durability testing noted a reduction in the transmittance of the masks that could be attributed the wear from the contact forces. As evident in the ultrastructural analysis, the PETG surface appears to have several accretions and disruption of the uniform coated surfaces. Given these PPEs are disposable and have a limited time of use, this may not significantly impact the PPE performance. In fact, this lack of clarity from routine use may be a good marker for replacement with a simple optical interferometry device.

This study has a few limitations. First, the biological performance of this anti-fogging solution could be examined using *in vitro* (reconstructed human epidermis), and *in vivo* skin contact testing in animals and human studies (*Arnesdotter et al., 2021*; *Choksi et al., 2019*; *Filaire et al., 2022*; *Hardwick et al., 2020*; *Nabarretti et al., 2022*; *Pistollato et al., 2021*; *Riebeling, Luch & Tralau, 2018*). While this could be considered routine diligence, it is prudent to emphasize both the waxes and PETG are generally regarded as safe (GRAS) and are used in current product (*e.g.*, lip balm) formulations routinely. The solvents are readily eliminated by mild heating (40 °C overnight). Second, applying a hydrophobic solution on the PETG clear surface may, counter to its primary objective, worsen the visibility due to the moisture coalescing into larger water beads. It is conceivable that the increasing size of these water beads may eventually reach a critical mass and cause them to slide off. This phenomenon is observed on hydrophobic surfaces like lotus leaves and commercial car wax coatings. Hence, a replaceable absorbent liner and a mild, mechanical vibration module to promote water droplet beading and run-off could be attractive future design iteration in these clear PPEs. Third, a major shortcoming of using such an anti-fogging spray is its temporary nature and the need for continuous replacement throughout the lifetime of the clear PPE. Thus, a potential future possibility is to pursue nanopatterning, simulating the lotus leaf papillae for a more biocompatible, permanent anti-fogging solution.

## CONCLUSIONS

In summary, this work demonstrates the utility of an anti-fogging formulation with two natural waxes that has good durability and optimal non-fouling properties. This solution represents a cost-effective, durable, non-toxic approach to prevent or reduce fogging of clear PPEs such as plastic masks and face shields from condensations of humidity arising from breathing and speaking.

## ACKNOWLEDGEMENTS

We thank the members of the Buffalo3DPPE for their time and effort during the pandemic namely, Kierra Bleyle, Preethi Singh, Jason Ciano, Savannah Tomaka, Jacob Graca, Hunter Rosa, Yianni Savidis, Danielle Detwiler, Omer Hillel, and Georgia Kyriacou. We also thank the Buffalo Enable (BE Mask) team for their valuable collaboration and technical assistance with 3D printing namely Aaron Gorsline, Albert Titus, James Whitlock, Peter Elkin, Jeremy Simon, Jon Schull, Ben Rubin, Kelly Cheatle, Pete Suffoletto, and Skip

Meetze. We would like to thank Mr. Peter Bush, South Campus Instrument core for assisting with the SEM analysis and Weber Medical laser for providing their laser device.

### Funding

This work was supported by BlueSky, University at Buffalo, and the Dean's fund, School of Dental Medicine. The funders had no role in study design, data collection and analysis, decision to publish, or preparation of the manuscript.

### Grant Disclosures

The following grant information was disclosed by the authors:
BlueSky, University at Buffalo.
Dean's Fund, School of Dental Medicine.

### Competing Interests

Praveen Arany is an Academic Editor for PeerJ.

### Author Contributions

- Succhay Gadhar conceived and designed the experiments, performed the experiments, analyzed the data, performed the computation work, prepared figures and/or tables, authored or reviewed drafts of the article, and approved the final draft.
- Shaina Chechang performed the experiments, prepared figures and/or tables, authored or reviewed drafts of the article, and approved the final draft.
- Philip Sales performed the experiments, authored or reviewed drafts of the article, and approved the final draft.
- Praveen Arany conceived and designed the experiments, analyzed the data, prepared figures and/or tables, authored or reviewed drafts of the article, and approved the final draft.

### Data Availability

  The raw data is available in the Supplemental Files.

### Supplemental Information

Supplemental information for this article can be found online at http://dx.doi.org/10.7717/peerj-matsci.30#supplemental-information.

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
