# Peer review of "Durability and physical characterization of anti-fogging solution for 3D-printed clear masks and face shields"

_PeerJ Materials Science, doi:10.7717/peerj-matsci.30_

## Round 0.1 · original submission · Minor Revisions

Thank you for the submitted manuscript! Please address the comments of Reviewer 1 and Reviewer 2. The comments of Reviewer 1 can be addressed one-by one; Reviewer 2 has more general comments and some additions can be made to the introduction about the anti-fogging action mechanisms.

Reviewer 1 ·

Basic reporting

The article describes and characterizes a coating intended to increase the hydrophobicity of clear PPE, specifically face masks and shields. The coating is safe, non-toxic, and (authors may want to mention this) renewable/sustainable, compared to many hydrophobic coating technologies. There are some areas where wording is confusing and usage/grammar needs a tune-up (see section 4 for details) but overall, the writing is clear. Appropriate references are cited, figures are provided – though resolution on some of the images could be improved, and a diagram of the “optical clarity” setup is needed. Also, if any actual face shield/mask devices have been produced with this film technique, the paper could benefit from a sample image of one.

Experimental design

The research question is well-defined, to wit; can a simple, effective hydrophobic coating be generated with inexpensive, widely available natural products to improve the user experience of clear face masks/shields. I would like to see a bit more interrogation of the generated films from a chemical perspective – I think there is room to do this within the existing dataset (see detailed comments in section 4). The methods are sufficiently detailed for replication, though the sources for caranauba wax and beeswax should be identified. I do have concerns about the manuscript’s use of statistical methodologies on tiny datasets. T-testing a set of n=3 seems like overkill, especially when the results of different trials and permutations are very obviously significant and distinct. Though, it does raise the question – why such small sample sets? For samples that can be produced by a simple spray method on sheets of PETG, it should have been relatively trivial to make and test more. This small sample size may impact some of the conclusions in the paper (see section 3).

The assessment of “optical clarity” as described in the paper should be recontextualized or possibly have additional data. As far as I can tell, the film is placed directly on the sensor - this means the laser’s spot size, spread, scattering, and/or refraction (i.e. indicators of light quality) cannot be measured after passing through the film. What is being measured with this experimental setup is transmittance, I/I0. This would be a straightforward find/replace operation in that manuscript, but the authors note in the introduction that the ability of a patient to see and distinguish the covered part of a care provider’s face is important to the application (e.g. lip-reading by the hearing impaired).

Many articles on image quality/transparency of films show samples placed on top of text, a logo, or other image with clearly defined features and use this as a qualitative indicator of image quality. This would be a good supplemental piece of data that would be easy to produce.

Validity of the findings

The findings seem valid and reasonably significant throughout. Some of the conclusions drawn could use some tuning. One of the more glaring issues is how authors qualify the films as superhydrophobic (themselves defining superhydrophobicity as a film with a water contact angle > 150°), but only achieved a water contact angle of 150.3 ± 2.1°. The error bars clearly demonstrate a lack of statistical confidence in this claim that could only be resolved by higher n and thus smaller error. Considering the reliance on statistical treatments for the comparing samples to one another, this seems like a bit of a miss. Fortunately, it can be resolved simply by removing the term “superhydrophobic” throughout the manuscript and emphasizing the significant differences between the generated films as a result of solids ratio/loading and solvent identity. I do not feel that the significance, relevance, or usefulness of the findings is impacted at all by an arbitrary line at 150°.

The authors also introduce a variable in the experimental design (solids loading) that is not addressed in the results/discussion section. This should be considered when interpreting the results of the tests.

The remainder of my commentary will be a line-by-line analysis in section 4.

Additional comments

Line 55 – it is not clear what “impervious nature” of N95s is being referred to, or why this would affect patient anxiety. Please explain.
Line 59 – should be “differently abled” and “hearing-impaired”
Line 59 – is attenuation the correct term here? The wave amplitude isn’t going down, this is a (literal) signal-to-noise problem.
Line 95 – please explain in the manuscript why two natural products would contain Cu2O contaminants
Line 96 – this is a sentence fragment, needs a verb
Line 99 – why/how would a hydrophobic coating prevent waste? Please explain.
Line 100 – be more specific
Line 101 – may want to use “the present study” rather than “this study” to differentiate from the previous discussion of others’ work.
Lines 108-114 – the variation in the solids loading here is understated and needs to play into the discussion in the back half of the paper. 0.875, 0.875, 1, 1.25. Two variables are being changed, but only the ratio is really discussed.
Line 109 – specify probe vs bath
Line 123 – is this a good method for drop generation considering the use case?
Line 128 – “performed in triplicate” unless authors plan on adding higher n
Line 132 – see above discussion of optical quality vs transmittance
Line 139 – how was “constant manual pressure” assessed or controlled? If it wasn’t, remove “constant”
Line 150 – specify that the skin-equivalent surface is leather in this section, and specify whether the leather has been oil treated (and which side is exposed, i.e. rough or smooth).
Line 155 – ultrastructure seems to be a term specific to cell biology, and none of the images taken approach the relevant magnifications for it. It might be worth noting that the use of SEM here is for its topological/depth of field advantages as opposed to magnification – optical microscopy of these samples would not yield the detail needed
Line 158 – sputter coated with what (C, Au, Pt, etc.), what is the anticipated thickness at 120 seconds for the coater, what are the manufacturer and model of the sputter coater used
Line 159 – please add more detail about the image settings, working distance, imaging mode, etc. Also, the accelerating voltage here does not match the images, images were taken at 2 kV.
Line 165 – “Data was subjected to a Student’s t-test or …”
Line 166 – Could probably remove the statistical analysis entirely and these results would still be believable
Line 174 – Very bold claim for how close to the threshold the results are. Some of the films are superhydrophobic, some are not. Higher n/lower error is required to make this claim, but superhydrophobicity is not a required state for the paper to be valid. Relative comparisons of hydrophobicity will suffice.
Line 184 – “… routine plastic mask and face shield use”
Line 190 – “wettable”
Line 205 – the units being described here are pressures, not forces. Force would be area-independent. This can be resolved by calling it pressure, or normalizing the results to the tested contact area and reporting values in Newtons.
Line 209 – 17/150 is a difference of over 10%. It’s prima facie significant, p-values not required
Line 212 – ultrastructure mentioned again, go with “surface texture” or “topological” analysis
Line 215 – is this biweekly replacement schedule derived from the data? How? Which experiments?
Line 223 – “PPE”
Lines 228-229 – simplify, “focused on transparent PPE designs” and strike “as our innovative solution.”
Line 231 – “deterrent to” instead of “for”
Lines 233-235 – rewrite for clarity, I would recommend “PETG is a commonly-used thermoplastic with impact resistance, durability, ductility, chemical resistance properties well-suited to this application.” Also, authors mention PVC but I don’t see it mentioned anywhere else. Delete unless the PVC data will be added in a revision.
Lines 240-241 – simplify, “The acetone formulation was superior in generating the most hydrophobic product after spray coating.”
Line 243 – “utilize” would be better as “disperse the waxes via,” and strike “that resulted in an effective homogenous solution”
Line 245 – “were strongly hydrophobic”
Line 249-250 – Please discuss more about why the difference between methanol and acetone. Is it related to the dispersibility of the waxes? Surface compatibility with PETG? How much does the solids loading variation play into this (i.e. why does a nominally thicker coating at 1:2 perform worse than a thinner one)?
Lines 251-260 – the reason soap is most destructive to the films is that its amphiphilic nature allows it to form micelles around the (non-polar) waxes and thus solubilize them. I had assumed this was a negative control, but the discussion section focuses almost solely on this finding. In my opinion, the more interesting discussion is IPA vs water, considering that both caranauba wax and beeswax have substantial fractions of fatty alcohols. One might anticipate the IPA being a better solubilizing agent. More discussion, supported by observations, would be useful – is it drying faster? Does it smooth the films due to its better solubility/lower surface tension? These are the lines of inquiry I’d like to see explored here.
Line 265 – strike “a little,” and replace “that approximates light skin contact force” with “and was intended to approximate light skin contact”
Line 266 – “integrity” should be “hydrophobicity,” the SEM images show that the surfaces look completely different after wear testing. Integrity was likely affected.
Line 273 – could the simple optical interferometry device be the human eye? See commentary in section 2 on measuring optical clarity.
Line 275-276 – Much like the statistical analysis above, gating analysis by in vitro testing seems excessive. These are natural components on the FDA GRAS list – they’re even edible. Both are already productized as lip balm, so they’re fine for skin contact, even on mucus membranes. PETG is food safe and is already used in this PPE application. The solvents used for deposition can be readily eliminated by introducing an overnight baking/drying step at 40 C.
Line 276 – delete “super” from “hydrophobic”
Line 277-278 – instead of speculating, can’t this be tested?
Line 280 – change to “observed on hydrophobic surfaces like lotus leaves…”
Line 280 – why not just make the absorbent out of something biodegradable so it can be discarded with the mask? PPE is disposable by design
Line 281 – a powered vibration module is total overkill for something that is intentionally cheap and disposable. Would be disturbing to the end-users as well.
Line 284 – why go with something expensive and difficult like nanopatterning when you already have a feasible solution here? A daily or weekly spray-on coating is a much better solution than something that will drastically increase price and decrease yield.
Line 288 – “This solution represents”
Line 290 – “PPE such as plastic masks and face shields from condensation of humidity arising from breathing and speaking.”
Line 437 – “angel” should be “angle”

Figure 1 – include a detailed schematic for the laser setup, otherwise a larger/higher resolution image is needed. The reader can’t see where the film is. Consider adding an example of the optical clarity assessment mentioned in section 2.

Figure 2 – reorganize this into columns for readability. Column 1: 2B, 2D, 2F, Column 2: 2C, 2E, 2G. Figure is already correctly labeled with “transmission” as opposed to “optical clarity.”

Reviewer 2 ·

Basic reporting

Reporting is clear and concise with good usage of English.

Suggested improvements:
- Introduction should include more general mechanisms for anti-fogging agents (i.e., include diagrams, how each class of compounds works, etc.)
- Discussion should also include much more discussion of mechanisms of action for anti-fogging agents that were used -- i.e., discuss why certain ones worked better?

Experimental design

The skin contact apparatus is a nice example of good test development for a specific goal in mind. However, there should be some verification experiments to demonstrate why this apparatus is a good test method for the application.

Validity of the findings

Graphs may be better presented with confidence intervals to give a graphical idea of P values, similar to those shown in https://www.nature.com/articles/nmeth.2659

Additional comments

Good paper overall, but there are a lot of additions that need to be made in terms of discussion, background, and experimental signficance.

---

## Round 0.2 · Minor Revisions

Thank you! The comments of Reviewer 1 were addressed.

Pedro Silva, a Section Editor, has spotted a few very minor edits that you may wish to consider:

There are a few small issues to correct:

A) authors at several points use non-SI units (e.g. ºF or oz)

B) the photos in panel I of figure 5 seem to contradict the transmittance data of panels F/G. The transmittance data suggest that the methanol-based coatings have hardly any effect in optical transmittance, but the photo shows the opposite. This should be explained/discussed.

C) lines 204 ff. "The method of disinfection that showed most increase in light transmission was wipes alone while the most reduction was observed with Isopropyl Alcohol and paper towel method. These results suggest that using Isopropyl Alcohol with gauze is optimal for disinfection as it minimally affects antifogging coating properties after repeated use." The second sentence does not appear to logically follow from the first one :-(

D) lines 222 ff "Topological analysis noted the wear simulation resulted in significant accretions and scratches that likely contribute to the reduced light transmission observed. These results indicate the coating will likely need to be replaced over repeated PPE use as often as every other week" authors do not mention how they computed the coating-replacement frequency. This should be explained properly.

A reference to papers describing the basics of anti-fogging mechanisms (like 10.1016/j.jmrt.2023.01.015) might also be placed in the introduction, to help interested readers better understand the importance of key concepts measured by the authors (e.g. the importance of sliding angle and contact angle).

Reviewer 1 ·

Basic reporting

Authors have substantially revised the article in line with review feedback. No further issues, article is suitable for publication.

Experimental design

No further comments

Validity of the findings

No further comments

Additional comments

No further comments

---

## Round 0.3 · Minor Revisions

The authors addressed the comments of reviewers.Thank you for your hard work. There are a couple of technical issues, which can be sorted very quickly:

## Acknowledgements
Please remove all financial and grant disclosure information from the Acknowledgements. This information is required to appear only in the Funding Statement for publication:
- This work was supported by the BlueSky, University at Buffalo, and Deanís fund, School of Dental Medicine

This information should only be provided in the Funding statement [**here**](https://peerj.com/manuscripts/81084/declarations/#question_18).

## References
In the reference section, please provide the full author name lists for any references with 'et al.' including this reference: “ Razavi SMR OJ, Sett S, et al. .”[/FAIL][PASS]At the next revision, in the reference section, please provide the full author name lists for any references with 'et al.' including this reference: “ Razavi SMR OJ, Sett S, et al. .”

## Remove Figure Legends from Figure Files
Please remove all figure numbers, titles, and legends from the figure files. This information should not be contained in the figure file.

At the next revision, please provide replacement figures measuring minimum 900 pixels and maximum 3000 pixels on all sides, saved as PNG, EPS or vector PDF file format without excess white space around the images.

## Figure Quality
 Please reformat the y-axes of your bar graphs in Figure 4 so that the axes start at zero. Beginning the y-axis of a bar graph at a non-zero value can be misleading as bar graphs compare absolute values. Please do not add breaks in the y-axes to correct this issue.

Please provide replacement figures measuring minimum 900 pixels and maximum 3000 pixels on all sides, saved as PNG, EPS, or vector PDF file format without excess white space around the images [**here**](https://peerj.com/manuscripts/81084/files). #]

---

## Round 0.4 · Minor Revisions

Dear authors: Pedro Silva, the Section Editor, has commented and said:

"Authors have not yet addressed my question "The transmitance data suggest that the methanol-based coatings have hardly any effect in optical transmittance, but the photo shows the opposite." This should be explained/discussed".

In the same vein, there is also a disagreement between Figure 2 --- F, J and I sections. Thus, acetone-based samples look more transparent than the methanol-based ones, although they show a lower transmission on the corresponding graphs. It could be that the measurement method used by the authors does not correlate with the visuals. --- Some explanation is required.

---

## Round 0.5 · accepted · Accept

Thank you for doing the good work!